

# Effects of short-term variability of meteorological variables on soil temperature in permafrost regions

Christian Beer[1,2], Philipp Porada[1,2], Altug Ekici[1,3], and Matthias Brakebusch[1,2]

[1]Department of Environmental Science and Analytical Chemistry (ACES), Stockholm University, 10691 Stockholm, Sweden
[2]Bolin Centre for Climate Research, Stockholm University, 10691 Stockholm, Sweden
[3]Uni Research Climate, Bjerknes Centre for Climate Research, Bergen, Norway

*Correspondence to:* Christian Beer (christian.beer@aces.su.se)

**Abstract.** Effects of the short-term temporal variability of meteorological variables on soil temperature in northern high latitude regions have been investigated. For this, a process-oriented land surface model has been driven using an artificially manipulated climate dataset. Climate variability mainly impacts snow depth, and the thermal diffusivity of lichens and bryophytes. This latter effect
is of opposite direction in summer and winter in most regions. These impacts of climate variability on insulating surface layers together substantially alter the heat exchange between atmosphere and soil. As a result, soil temperature is 0.1 to 0.8 °C higher when climate variability is reduced. Earth system models project warming of the Arctic region but also increasing variability of meteorological variables and more often extreme meteorological events. Therefore, our results show that projected
future increases in permafrost temperature and active-layer thickness in response to climate change will be lower i) when taking into account future changes in short-term variability of meteorological variables, and ii) when representing dynamic snow and lichen and bryophyte functions in land surface models.

## 1 Introduction

Soil temperature is an important physical variable of a terrestrial ecosystem since it controls many functions of microbes and plants. In permafrost regions, soil temperature also defines the biologically active part of the soil that is thawing in summer (active layer). Therefore, impacts of future warming on soil temperature have been investigated in numerous experimental and modelling studies during the past decades. Large-scale soil temperature is mainly determined by vertical heat conduc-
tion. Therefore, soil temperature usually follows an annual sinusoidal cycle of air temperature with





a damped oscillation (Campbell and Norman, 1998). That is why the projected large increase in air temperature in the Arctic region over the next 100 years (Ciais et al., 2013) is raising large concerns about the response of soil temperature and hence permafrost thawing in the Arctic. Indeed, measurements during the last decades already show an increasing permafrost temperature (Romanovsky

et al., 2010) and active-layer thickness (Callaghan et al., 2010) in response to global warming. Also, first modelling results confirm such simple response of increasing future soil temperature and active-layer thickness (Schaefer et al., 2011; Koven et al., 2011; Lawrence et al., 2012; Peng et al., 2016). As a result of increasing soil temperature and active-layer thickness, heterotrophic respiration is suggested to increase because of the temperature-response of biochemical functions (Arrhenius, 1889;

van't Hoff, 1896; Lloyd and Taylor, 1994) and the additional availability of decomposable substrate (Schaphoff et al., 2013; Koven et al., 2015) potentially leading to a positive climate-carbon cycle feedback (Zimov et al., 2006; Beer, 2008; Heimann and Reichstein, 2008).

Meteorological variables, such as air temperature and precipitation will not only change gradually into the future but also their short-term variability and frequency of extreme events is projected

to change (Easterling et al., 2000; Rahmstorf and Coumou, 2011; Seneviratne et al., 2012). For instance, for northern high-latitude regions, climate models project an increase of the annual maximum of the daily maximum temperature by 4 °C by 2100 (Seneviratne et al., 2012) while annual maximal daily precipitation is projected to increase by 20% in these areas by 2100. At the same time, many ecosystem functions respond non-linearly to environmental factors, cf. for instance the temperature-

dependence of biochemical functions (Arrhenius, 1889). Therefore, effects of the short-term (daily to weekly) variability of meteorological variables on the long-term (decadal) mean ecosystem functions can enhance or dampen the effect of a general gradual warming (Reichstein et al., 2013; Schwalm et al., 2017). That is why there is a strong need to understand such effects of climate variability on ecosystem states and functions in addition to gradual changes in order to reliably project future

ecosystem state dynamics and climate. In this context, effects of climate variability on soil temperature in northern high latitude environments have not been studied so far: In addition to a gradual warming of Arctic air and soil temperature, what are the specific effects of changing short-term variability of meteorological variables on the long-term mean annual or seasonal soil temperature? Will a short-term variability change have the capability to enhance or dampen the anticipated soil

warming?

Due to the well-known dampening effects of snow, near-surface vegetation, and the organic layer (Yershov, 1998, pages 361-369) (Goodrich, 1982; Zhang, 2005; Wang et al., 2016; Jafarov and Schaefer, 2016), one would expect no to little additional effects of changing air temperature fluctuations on soil temperature, in particular not on subsoil and permafrost temperature. However, air

temperature variability will have an impact on snow height indirectly through snow density (Abels, 1892) and also directly when temperature is periodically rising above the melting point. In addition, the dependence of soil and near-surface vegetation conductivity on water and ice content (Camp-



bell and Norman, 1998) complicates the picture because water and ice contents themselves are also temperature-dependent. Snow manipulation experiments have proven the large *spatial* heterogeneity

of soil temperature in cold regions due to snow height heterogeneity (Wipf and Rixen, 2010). The temporal variability of insulating layers and their properties should be of similar importance for soil temperature.

At high latitudes, near-surface vegetation consists to a large part of lichens and bryophytes, which often form a continuous layer on the ground. Lichens are symbiotic organisms consisting of a fungus

and at least one green alga or cyanobacterium, while bryophytes are non-vascular plants which have no specialised tissue such as roots or stems. Both groups cannot actively control their water uptake and loss, but they tolerate drying and are able to reactivate their metabolism on rewetting. Typical species of upland regions at high latitudes are feather mosses such as *Hylocomium splendens* and *Pleurozium schreberi* or the lichen *Cladonia stellaris*. This near-surface vegetation is growing on

top of any organic horizon and hence important for heat fluxes between land and atmosphere. In particular also for this layer, thermal and hydrological properties depend highly on water and ice content. Hence, lichens and bryophytes dynamically influence the vertical heat conduction (Porada et al., 2016a).

This study investigates the effects of *temporal* variability of meteorological variables on snow and

lichen/bryophyte insulating properties and hence soil temperature in permafrost regions. For this, a recently advanced land surface model (LSM) has been used that also represents permafrost-specific processes, and in particular a dynamic snow representation and a dynamic near-surface vegetation model (Porada et al., 2016a). While the model has been evaluated against several types of observations in other studies (Ekici et al., 2014, 2015; Porada et al., 2016a; Chadburn et al., 2017), here

mean annual ground temperature (MAGT) is evaluated again against different observations or other modelling studies. Then, the model is run with two distinct climate forcing datasets, one control dataset and one that has identical long-term averages but reduced day-to-day variability of meteorological variables, such as air temperature and precipitation. The differences in long-term average results from these two model runs will therefore demonstrate the exclusive effects of temporal vari-

ability of climate variables and extreme meteorological events on MAGT in high latitude permafrost regions.

## 2    Methods

### 2.1    The land surface model JSBACH

The Jena Scheme for Biosphere-Atmosphere Coupling in Hamburg (JSBACH) is the land surface

scheme for the Max Planck Institute Earth System Model (MPI-ESM) (Raddatz et al., 2007; Reick et al., 2013). It runs coupled to the atmosphere inside the ESM or offline forced by observation-based or projected climate input data. This model has recently been advanced by several processes which



are particularly important in cold regions (Ekici et al., 2014): coupling of soil hydrology and heat
conduction via latent heat of fusion and the effects of soil ice and water content on thermal properties,
and a snow model for soil insulation. The model simulates heat conduction and soil hydrology in
a 1-D vertical scheme using several layers (Hagemann and Stacke, 2015). The version used in this
study has been updated from the one used in Ekici et al. (2014) by two additional deep soil layers for
thermal and hydrological processes of 13 and 30 m, respectively, which lead to a total potential soil
profile of 53 m. However, soil hydrological processes are constrained by the depth to the bedrock.
Another constraint on soil hydrological processes is the potentially available pore volume which is
reduced by ice content.

In contrast to the model version described in Ekici et al. (2014), here we use a further advanced
snow module that includes *dynamic* snow density and snow thermal properties (Ekici, 2015). In
this approach, the snow density ($\rho_{snow}$) follows a similar representation as in Verseghy (1991). It is
initialized with a minimum value of $\rho_{min} = 50 kg m^{-3}$. Then the compaction effect is included as a
function of time and a maximum density ($\rho_{max} = 300 kg m^{-3}$) value (Eq. 1),

$$\rho_{snow}^{t+1} = \left( \rho_{snow}^{t} - \rho_{max} \right) \exp \frac{-0.002 \cdot \Delta t}{3600} + \rho_{max} \qquad (1)$$

where $\Delta t$ is the timestep length of model simulation. Additionally, when there is new snowfall, snow
density is updated by taking a weighted average of fresh snow density ($\rho_{min}$) and the calculated
snow density value of the previous timestep.

Snow density controls snow heat conduction parameters. Eq. 2 and Eq. 3 show the relationships
of volumetric snow heat capacity ($c_{snow}$) and snow heat conductivity ($\lambda_{snow}$) to snow density fol-
lowing the approach of Abels (1892) and Goodrich (1982). With no previous snow layers, $c_{snow}$ is
initialized with an average value of $0.52 MJ m^{-3} K^{-1}$ and $\lambda_{snow}$ with $0.1 W m^{-1} K^{-1}$,

$$c_{snow} = c_{ice} \cdot \rho_{snow} \qquad (2)$$

where $c_{ice}$ is the specific heat capacity of ice $\left( 2106 J kg^{-1} K^{-1} \right)$, and

$$\lambda_{snow} = 2.9 \cdot 10^{-6} \cdot \left( \rho_{snow} \right)^2 \qquad (3)$$

Another important advancement of the JSBACH model version used in this study is the inclusion
of a dynamic lichen and bryophyte model (Porada et al., 2013, 2016a). This model is designed to
predict lichen and bryophyte net primary productivity (NPP) in a process-based way from available
light, surface temperature, atmospheric carbon dioxide concentration, and water content of lichens
and bryophytes. Furthermore, it is applicable to estimate various impacts of lichens and bryophytes
on biogeochemical cycles (Porada et al., 2016b; Lenton et al., 2016; Porada et al., 2017). The model
includes a dynamic representation of the surface cover which depends on the balance of growth due
to NPP and reduction by disturbance, such as fire (Porada et al., 2016a). The coverage of the layer
determines its influence on heat exchange between atmosphere and soil. The layer thickness and
porosity is set to 4.5 cm and 80%, respectively.



The lichen and bryophyte water balance is integrated into the scheme of hydrological fluxes in JSBACH. In addition, the lichen and bryophyte layer is fully integrated into the heat conduction

scheme and hence also functions as a soil insulating layer (Porada et al., 2016a). Soil insulation depends on the fractional grid cell coverage of the lichen and bryophyte layer as well as on its hydrological status. Thereby, thermal diffusivity of this layer is computed as a function of water, ice and air content in the lichen and bryophyte layer (Porada et al., 2016a). The simulated relations between thermal properties of the lichen and bryophyte layer and water content agree well with

field observations. Porada et al. (2016a) provide a complete description of the dynamic lichen and bryophyte model in JSBACH. The model version used here differs from Porada et al. (2016a) only with respect to the parametrisation of the snow layer, which has a slightly longer compression time, and a few bug fixes. This updated version is also used in Chadburn et al. (2017), where it shows good agreement with site level soil temperature observations.

**2.2   Model experiments**

For addressing the research question about effects of climate variability on mean annual ground temperature in permafrost regions (cf. section 1), artificial model experiments are conducted in this study. In addition to the control model run (CNTL), in one model experiment called REDVAR the land surface model has been driven by an artificial climate dataset that represents a reduced short-

term (day-to-day) climate variability while the decadal averages are conserved (section 2.4). Then, differences in decadal averages of simulated snow and lichen and bryophyte properties and ultimately soil temperature can be interpreted exclusively due to a difference in variability of meteorological variables.

Two different kinds of such experiments are presented in this study. The main experiments are

conducted at the pan-Arctic scale over historical to recent time periods (1901-2010). Here, CNTL and REDVAR model runs are done exactly the same way including the spin-up approach for reservoir initialization. At the end, results are compared from "two different worlds" with the same average climate, one with a constantly lower variability of meteorological variables than the other.

The second kind of experiments has been performed at site-level scale. Here, JSBACH has been

run over the period 1901-2100 (CNTL) and a second model run with constantly *increasing* reduction of climate variability (REDVARfut, see section 2.4) has been performed for the period 2011-2100. This experiment additionally clarifies the effects of changing future climate variability on permafrost temperature. The REDVARfut experiment additionally contribute to the question on how climate data should be prepared in order to perform so called offline model experiments into the future.

Of particular concern are potential biases in future projections of ecosystems states using LSMs because in these projections anomalies of raw ESM output is usually added to recent short-term variability of meteorological variables. Even if that is the most reliable approach of conducting such future projections at the moment, still we need to address the question, how high could be the bias



just because a change in short-term variability has been neglected? The REDVARfut experiment has
been conducted for two grid cells representing two sites, one Canadian site at about 62.2N, -75.6E
with MAGT of about -5 deg C, and one East Siberian site at about 72.2N, 147E with MAGT of about
-10 deg C. At these sites, JSBACH results differed by only 0.7 and 0.2 deg C from these borehole
measurements.

State variables have been brought into equilibrium using a spin-up approach prior to the transient
model runs (1901-2010 or 1901-2100). We assume the time period 1901-1930 to be a representative
for pre-industrial climatology following (Cramer et al., 1999; McGuire et al., 2001). Therefore,
randomly selected years from that period have been used. For a proper spin-up of soil physical
state variables in permafrost regions, we suggest a 2-step procedure. First, a 50-year model run with
the above described randomly selected climate from the period 1901-1930 has been done without
considering any freezing and thawing. This first spin-up will bring the soil temperature and water
pools in a first equilibrium with pre-industrial climate. In a second step, another 100 years spin-up
with the same climate data is performed but now freezing and thawing is switched on in order to
have all pools including soil ice and water content, and soil temperature in equilibrium with climate.

### 2.3 Forcing data

The JSBACH model estimates half-hourly climate forcing data using daily data of maximum and
minimum air temperature, precipitation, short-wave and long-wave radiation, specific humidity and
surface pressure. We are using global data at 0.5 degree spatial resolution which has been produced
following the description in (Beer et al., 2014). The historical data from 1901-1978 came from the
WATCH forcing dataset (Weedon et al., 2011), and for the period 1979-2010 ECMWF ERA-Interim
reanalysis data (Dee et al., 2011) has been bias-corrected against the WATCH forcing data following
Piani et al. (2010) as described in Beer et al. (2014).

For a specific additional projection into the future (REDVARfut, section 2.2), meteorological data
during 2011-2100 have been obtained from the CMIP5 output of the Max-Planck-Institute Earth
System Model (Giorgetta et al., 2012) following the representative concentration pathway (RCP)
8.5. Meteorological data of the two grid cells representing the Canadian and Russian sites were cut
out and then also bias corrected to the observation-based period following Piani et al. (2010) as
described in Beer et al. (2014).

Grid cells are divided into four tiles according to the four most dominant vascular plant func-
tional types of this grid cell (Ekici et al., 2014). This vascular vegetation coverage is assumed to
stay constant over the time of simulation. In the model simulations used in this study, we apply
new soil parameters. Hydrological parameters have been assigned to each soil texture class follow-
ing Hagemann and Stacke (2015) according to the percentage of sand, silt and clay at 1 km spatial
resolution as indicated by the Harmonized World Soil Database (FAO/IIASA/ISRIC/ISSCAS/JRC,
2012). Thermal parameters have been estimated as in (Ekici et al., 2014) at the 1 km spatial resolu-



tion. Then, averages of 0.5-degree grid cells have been calculated. Soil depth until bedrock follows
the map used in Carvalhais et al. (2014) based on Webb et al. (2000).

### 2.4   Meteorological forcing data with manipulated variability

Based on the climate data described above (subsequently called CNTL dataset), an additional cli-
mate dataset has been developed. This dataset shows reduced day-to-day variability but conserved
long-term mean values when comparing to CNTL, as described in detail in Beer et al. (2014). The
dataset with reduced variability is called REDVAR. In that dataset, the variability of daily values is
reduced by a variance factor of $k = 0.25$ (see Beer et al. (2014) for details), but the mean seasonal cy-
cle is conserved. The seasonal variability is represented by an 11-year running average across same
dates. Differently from Beer et al. (2014), seasonal means in the REDVAR dataset were exactly pre-
served by normalization with respect to the CNTL dataset for the annual quarters December-January-
February, March-April-May, June-July-August, and September-October-November for each year in-
dividually.

For the specific additional projection until 2100 at site-level scale, bias-corrected future climate
data has been manipulated such that the short-term variability of meteorological variables *is dynam-*
*ically* reducing during 2011-2100, in contrast to the REDVAR dataset for which a constant reduction
factor has been applied. This additional artificial dataset is called REDVARfut in the following.
For REDVARfut, the variance factor $k$ is set to change linearly from 1 to 0.1 over these 90 years
following Eq. 4:

$$k = 1 - \left(2.7^{-5} \cdot d\right) \tag{4}$$

where $d$ is the day relative to 1 Jan 2011. This has been done for two grid cells representing one
location in Canada (medium recent MAGT) and one location in East Siberia (cold recent MAGT)
(cf. section 2.2). The CNTL and REDVARfut datasets are identical for the time period before 2011.

## 3   Mean annual ground temperature evaluation

The frost-enhanced JSBACH model has been intensively evaluated elsewhere (Ekici et al., 2014,
2015; Porada et al., 2016a). The model version used here has also been recently extensively evalu-
ated against site-level observations (Chadburn et al., 2017). In this paper, the simulated mean annual
ground temperature (MAGT) is again evaluated against various other datasets at different spatial
scales. First, JSBACH model results are compared to model results from the GIPL 1.3 model (Uni-
versity of Alaska Fairbanks) over Alaska for the period 1980-1989. For this comparison we used
JSBACH mean soil temperature results from layer 7 (38 m depth) and during 1980-1989. Then, spa-
tial details of MAGT are compared to the information from the Geocryological Map of Yakutia (Beer
et al., 2013) using also model results from layer 7 but a mean value during 1960-1989. The depth
of 38 m ensures that temperature variation is negligible and hence comparable to the information in



the observation-based map. Last, JSBACH subsoil temperature is compared to pan-Arctic borehole
measurements collected by the GTN-P initiative (Romanovsky et al., 2010; Christiansen et al., 2010;
Smith et al., 2010) using model results from the layer corresponding to the measurement depth and
from year 2008. The respective GTN-P Thermal State of Permafrost (TSP) snapshot data has been
dowloaded from the National Snow and Ice Data Center (NSIDC).

### 3.1   Analysis

In order to analyse effects of variability of meteorological variables on snow and near-surface veg-
etation properties and hence soil temperature, model results have been averaged during the period
1980-2009. As the averages of climate forcing data is similar between both experiments REDVAR
and CNTL, (relative) differences in long-term average model results, such as snow depth or soil tem-
perature, show the effects of short-term variability of climate forcing data. Relative differences are
displayed as a fraction (no unit). In Fig. 4 to Fig. 9 the dark green area represents all land outside the
(sporadic) permafrost zone which is masked by applying a long-term mean air temperature threshold
of -3 °C .

## 4   Results

### 4.1   Mean annual ground temperature evaluation

When comparing against a global dataset of mean annual ground temperature (MAGT) at depth
ranging usually from 1 to 20 m (GTN-P initiative) JSBACH shows almost no bias (-0.4 °C ) and
a root mean square error of 3 °C Fig. 1. JSBACH represents the spatial variation in mean annual
ground temperature (MAGT) reasonably well with a coefficient of determination of 0.5. Fig. 1 shows
that for a number of measurements between 0 and -1 °C , JSBACH simulates a larger variation
ranging from 2 to -8 °C . In addition, JSBACH clearly underestimates MAGT at three borehole
sites in the Canadian High Arctic (data about -10 °C , model about -22 °C ) which requires further
evaluation, e.g. about the representativeness of these data points or about the validity of snowfall
input data to the model.

When looking at alternative estimates of spatial details of MAGT, JSBACH both underestimate
or overestimate MAGT by about 2 to 4 °C depending on the location (Fig. 2,Fig. 3). The JSBACH
results for Alaska are compared to another model output. JSBACH overestimates MAGT in many ar-
eas in Alaska by several °C while also underestimates MAGT at the southern end of the North Slope
(Fig. 2). In East Siberia (Yakutia), the model usually underestimates MAGT by 2 to 6 °C (Fig. 3)
when comparing to an observation-based map (Beer et al., 2013). However, the cold bias is largely
reduced when taking the uncertainty (standard deviation) in the original geocryological map into ac-
count (Fig. 3). Then, the difference is negligible in many regions. Still, there is a very strong cold bias
in the mountainous regions of East Siberia. When taking the map uncertainty into account (Fig. 3)



the model still underestimates MAGT by about 6 to 8 °C here. This bias can also not be explained by the general warm bias of very low MAGT in the geocryological map when comparing to GTN-P

observations (Beer et al., 2013). In fact, very low snow depth model results in these areas of about 15 cm on average (data not shown) seem to be the reason for a too low insulation of soil during a very cold winter.

### 4.2 Climate forcing data comparison

The long-term (1980-2010) averages of air temperature differ by only 0.015 °C at maximum or

0.004 % between CNTL and REDVAR in permafrost regions (Fig. 4a). Also long-term precipitation averages are similar between the datasets, with differences of -0.2 to 0.1 % (Fig. 4b).

In contrast, the difference in short-term variability of meteorological variables at daily resolution between both datasets is remarkable. Although the statistical transformation of variables has been performed at residuals to the mean seasonal cycle (section 2.4), still the standard deviation of air

temperature at daily resolution is usually 0.2 to 1 °C lower in the REDVAR dataset compared to CNTL, or 2 to 10 % (Fig. 5a). That means that temperature of warmer days have been reduced while air temperature of colder days have been increased such that the overal mean air temperature is similar. Interestingly, the amount of variability difference between the two datasets also depends on the location. For example, lower standard deviation differences are visible towards colder regions,

such as East Siberia and the Canadian High Arctic. One explanation for this pattern is the higher mean seasonal cycle in continental climate, which has not been manipulated (section 2.4), and which therefore dominates stronger the overall variability, which is analyzed in Fig. 5a. Also REDVAR precipitation standard deviation is usually 2 to 6 % lower than precipitation standard deviation of the CNTL dataset (Fig. 5b).Hence, in this artificial climate dataset, extremely heavy rainfall or snowfall

is reduced while small precipitation amounts have been increased.

### 4.3 Climate variability effects on snow properties

Importantly, snow depth is up to 20 percent higher under reduced climate variability conditions (Fig. 6a). In fact, the snow depth difference can be explained by differences in snow water equivalents of same magnitude (Fig. 6b). In contrast, the slightly higher snow density under reduced climate

variability (Fig. 6c) is not able to explain the difference in snow depth. Snow melt flux differences in autumn between both model experiments of 10 to 40 percent (Fig. 6d) demonstrate clearly that under reduced air temperature variability during the beginning of the snow season, individual snow melt events and hence the total snow melt flux are reduced. Besides snow depth, the thermal diffusivity of snow controls the overall heat conduction. Fig. 7 shows that under reduced climate variability

conditions, thermal diffusivity of snow is 0.5 to 2.5 percent higher in high latitude regions.



### 4.4 Climate variability effects on thermal diffusivity of lichens and bryophytes

Thermal diffusivity of lichens and bryophytes differs only marginally between the REDVAR and CNTL model experiments over most of the northern high latitude permafrost regions (Fig. 8a). In western Siberia and Quebec, winter moss thermal diffusivity is up to 12 percent lower under reduced
climate variability conditions (Fig. 8a). In contrast, summer moss diffusivity is usually higher under reduced variability of meteorological variables (Fig. 8b). Under these climate conditions, it is raining more often a little bit and air temperature are not extreme resulting in more moist conditions for lichens and bryophytes, hence higher thermal diffusivity. In tundra the difference is about 2 percent while in the boreal forest it can be up to 6 percent (Fig. 8b).

### 4.5 Ultimate climate variability effects on soil temperature

The estimated long-term average of both topsoil and subsoil temperature differs between REDVAR and CNTL experiments (Fig. 9a,Fig. 9b). Soil is 0.1 to 0.8 °C warmer when climate variability is reduced (Fig. 9a,Fig. 9b). These results and also the spatial pattern are similar between topsoil and subsoil values (Fig. 9a,Fig. 9b) with a bit larger effect on topsoil temperature. Soil temperature
differences are larger in winter with values up to 1.5 °C compared to the summer when differences are typically 0.2-0.5 °C (Fig. 9c,Fig. 9d).

### 4.6 Effects of future changes of climate variability on soil temperature

In order to analyze effects of changing variability of meteorological variables in time, the results of the respective additional model runs into the future at two sites are displayed as time series in
Fig. 10 and Fig. 11. In contrast to the continental model experiments, in these additional point simulations the variability of meteorological variables is increasingly reduced during 2011-2100 in the REDVARfut input dataset while the historical climate until 2010 is identical (section 2.4).

The bias-corrected MPI-ESM CMIP5 model output following RCP8.5 shows increasing air temperature in both locations (solid blue line in Fig. 10a and Fig. 11a). Precipitation is also slightly in-
creasing (solid blue line in Fig. 10b and Fig. 11b). This positive trend is also seen by the annual minimum (percentile 1) and maximum (percentile 99) temperature, and maximum precipitation (dashed blue lines in Fig. 10 and Fig. 11). Meteorological forcing data of the REDVARfut dataset (red lines) shows similar long-term averages to the CNTL dataset (Fig. 10a,Fig. 10b,Fig. 11a,Fig. 11b). Hence, REDVARfut variables follow the general positive trend. However, as the short-term variability is de-
signed to being increasingly reduced, the differences in the minimum (1-percentile) and maximum (99-percentile) air temperature are increasing during 2011-2100. The increasing maximum daily precipitation in the CNTL dataset has been reversed in REDVARfut where the amount of precipitation at percentile 99 is even decreasing in time (Fig. 10b, Fig. 11b).





These CNTL and REDVARfut climate datasets have been used as forcing data for JSBACH in the
additional point-scale model runs. The respective soil temperature results are compared to each other
in Fig. 10 and Fig. 11. The time-varying changes in the variability of meteorological variables under
conserved long-term average leads to a difference in topsoil temperature of up to 0.8 °C (Fig. 10c,
Fig. 11c), i.e. the overall increasing topsoil temperature due to increasing air temperature is a bit
higher in case of reduced climate variability. This effect is also visible in 38 m depth (Fig. 10d,
Fig. 11d) even though short-term atmospheric data fluctuations should be most filtered at this depth.

## 5   Discussion

Climate model projections show increasing variability of meteorological variables and hence in-
creasing frequency of extreme meteorological events (Seneviratne et al., 2012) along with a gradu-
ally changing climate (change of long-term mean values) (Ciais et al., 2013). Because of the non-
linearity of ecosystem response functions, changing extreme event frequency and changing variabil-
ity of meteorological variables can have a higher impact on ecosystem state and function than a
gradual change of mean meteorological variables (Reichstein et al., 2013; Beer et al., 2014). This
study contributes to this overall question from a theoretical point of view with LSM experiments for
which artificially manipulated climate forcing datasets have been employed. These climate datasets
practically do not differ in their decadal averages (section 4.2) while they are showing a substan-
tial difference in the short-term (daily) variability (section 4.2). Therefore, differences in simulated
state variables and fluxes over 30-year periods (soil temperature in this case) will be only due to
differences in *temporal variability* of meteorological variables. This study addresses particularly the
question about the effect of climate variability on soil temperature in northern high latitude regions.
The CNTL experiment shows *higher* climate variability than the artificial experimental REDVAR
dataset (sections 2.4 and 4.2), and respective model result differences between experiments using
the manipulated climate REDVAR) and the CNTL dataset are shown in section 4. Methodologi-
cally, it is important to artificially design a climate dataset with *reduced* temporal variability because
otherwise there is a high risk for producing a physically unrealistic climate conditions. However,
for interpreting the results in terms of future ecosystem responses to *increasing* climate variability
(Seneviratne et al., 2012), **the direction of the conclusions are carefully inverted in this discus-
sion section**.

In contrast to the climate forcing data, the long-term average of both topsoil and subsoil tem-
perature differs between REDVAR and CNTL experiments (Fig. 9a, Fig. 9b). The same is true for
respective future projections (Fig. 10, Fig. 11). In fact, under higher variability of meteorological
variables and higher frequency of extreme events (CNTL versus REDVAR experiments) soil will be
cooler (Fig. 9c, Fig. 9d, Fig. 10, Fig. 11) given all other environmental factors are similar. That means
that the projected increase in future variability of meteorological variables (Seneviratne et al., 2012)



has the potential to dampen soil warming occurring as a function of increasing mean air temperature.

To further understand the underlying processes, individual effects of climate variability on snow and near-surface vegetation properties are discussed in the following paragraphs.

For land-atmosphere heat conduction the thermal properties of snow, near-surface vegetation (e.g. mosses and lichens), the soil organic layer, and their spatial extent and heights are of major importance (Yershov, 1998; Gouttevin et al., 2012; Wang et al., 2016; Jafarov and Schaefer,

2016). Snow generally insulates the soil from changing atmospheric temperature. However, effects are smaller during the melting period in spring because the snow is wet and conductivity therefore higher, and more importantly, the soil-to-air gradient in temperature is small. The insulation effect of near-surface vegetation also differs among the seasons because of the high dependence of thermal properties on water and ice contents of lichens and bryophytes. Usually, dry lichens and bryophytes

during a continental summer should insulate much more than during wet spring or autumn, or during the ice-rich winter time.

This theoretical study shows that one major effect of higher climate variability on cold region environments is a lower snow water equivalent (section 4.3) which directly translates into lower snow depth values. The potential alternative explanation for a lower snow depth would be a higher snow

density. However, the results show exactly the opposite (Fig. 6c). In addition to snow depth, snow thermal properties are also an important factor for heat conduction. However, winter snow thermal diffusivity is some percent lower under higher climate variability conditions (CNTL-REDVAR). Therefore, the net *snow-related* effect of higher climate variability on soil temperature, that is a cooler soil (section 4.5) is explained by snow depth differences alone, i.e. a lower snow depth under

higher climate variability.

The reason for these snow water equivalent differences are more often circumstances of melting snow during the beginning of the snow season when day-to-day variability of air temperature is higher (section 4.3). These results also point to an interesting combination of impacts of both changing variability *and* gradually changing mean values on ecosystem states because both changes can

lead to pass a threshold value (melting point in this case). These impacts can be seen in section 4.3 when combining temporal climate variability effects on snow water equivalent results (Fig. 6) and snow melt flux results (Fig. 6d) with longitudinal pattern of these results towards a continental climate, which can be interpreted in terms of gradual climate change when substituting space for time. Overall, these findings show that projected higher climate variability in future can lead to lower snow

depth which will reduce a soil warming in response to air warming.

In addition to the insulating effect of snow, lichens and bryophytes growing on the ground influence on heat conduction (Porada et al., 2016a). It is interesting to note that when climate variability is higher (CNTL conditions), moss thermal diffusivity can be substantially *higher in winter* and *lower in summer* in the same region (section 4.4). This fact points to an important role of near-surface

vegetation: it will insulate less from air temperature during winter and insulate more during summer



with increasing climate variability in future. These effects of climate variability on thermal diffusivity of lichens and bryophytes and hence soil temperature are in the same direction as snow effects (section 4.3), again reducing the soil warming effect of future climate change.

Effects of climate variability on both snow and moss properties are in the same direction (sections 4.3 and 4.4). As a result, soil will be cooler under higher climate variability (section 4.5). Recent modelling studies suggest a soil temperature increase of 0.02 °C per year since 1960 (McGuire et al., 2016) which translates into 2 °C in 100 years. Such soil temperature increase has also been projected using the JSBACH model under the RCP4.5 scenario (Ekici, 2015) while under the strong warming scenario RCP8.5, the soil temperature increase might be up to 6 to 8 °C (Ekici, 2015). Lower soil temperature under higher climate variability in the range 0.1 to 0.8 °C (section 4.5) demonstrate that under increasing variability of meteorological variables and increasing extreme events in the Arctic (Seneviratne et al., 2012), the effect of gradual air temperature increase on soil temperature and hence active-layer thickness will be *dampened*. Such dampening of future soil warming will also reduce the otherwise positive biogeochemical feedback to climate (Zimov et al., 2006; Beer, 2008; Heimann and Reichstein, 2008). Our results are conservative here because the 99 percentiles of air temperature and precipitation from the artificial dataset (REDVAR) differ by only 1-4 °C (temperature) and 1-10 % (precipitation). These values are at the lower end of the range of climate model projections for the Arctic region until 2100 (Seneviratne et al., 2012).

The presented effects of short-term variability of meteorological variables on ecosystem states and functions, such as soil temperature, are also important from a methodological point of view. To study the effects of environmental change on ecosystems, LSMs are usually forced by historical and reanalysis climate data for the past and present periods, and by future climate results from Earth system models. Since ESM results usually show biases, the ESM outputs cannot be used directly to drive the LSM offline model runs but first need to be bias-corrected (Hempel et al., 2013). The results of the presented REDVAR and REDVARfut experiments demonstrate that such bias-correction methods should account for the projected change in short-term (daily) variability in addition to general trends.

In addition, a first run of the MPI-ESM with the permafrost-advanced land surface scheme JS-BACH coupled to the atmosphere model showed a remarkable bias in 2m air temperature of 1-4 °C in permafrost regions compared to the standard model version without freezing and thawing (Hagemann et al., 2016). Our results suggest that this bias could be potentially reduced when implementing representations of dynamic snow and of dynamic lichens and bryophytes.

Our findings have three major implications for future permafrost science:

1. New highly controlled laboratory and field experiments are required in order to confirm modelling results about climate variability effects on permafrost soil temperature.

2. Future developments of land surface models should include dynamic models of snow, and lichens and bryophytes.




3.  Statistical methods need to be developed such that future forcing data for climate change
    impact studies can be prepared in a way that a potential change in short-term variability and
frequency of extreme events is preserved.

## 6   Conclusions

Artificial model experiments have been used in order to quantify the impact of the variability of
meteorological variables on the long-term mean of mean annual ground temperature in permafrost-
affected terrestrial ecosystems. This impact is mainly due to temperature variability effects on snow
melt and snow depth as well as climate variability effects on the (seasonally different) thermal dif-
fusivity of lichens and bryophytes. Overall, the soil temperature response to increasing climate vari-
ability and extreme event frequency (soil cooling) will be opposite to the response of soil temperature
to gradually increasing air temperature (soil warming). This shows the importance of representing
dynamically snow and lichen and bryophyte functions in Earth system models for projecting fu-
ture permafrost soil states and land-atmosphere interactions, hence future climate. Our findings also
point to the need to represent changes in short-term variability of meteorological variables in bias-
corrected climate data of future periods.

*Acknowledgements.* Financial support came from the European Union FP7-ENV project PAGE21 under con-
tract number GA282700. Model simulations were performed on resources provided by the Swedish National
Infrastructure for Computing (SNIC) at Linköping University. We acknowledge the Land Department, Max
Planck Institute for Meteorology, Hamburg, Germany for JSBACH code maintenance. We thank Charles Koven
for a constructive review that helped to improve a previous version of the manuscript. We further acknowledge
the borehole temperature dataset "IPA-IPY Thermal State of Permafrost (TSP) Snapshot Borehole Inventory,
Version 1.0" downloaded from NSIDC.





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





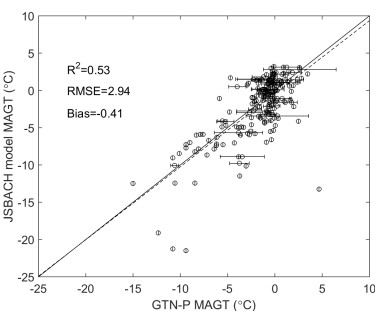

Figure 1: Evaluation of mean annual ground temperature against GTN-P borehole measurements.

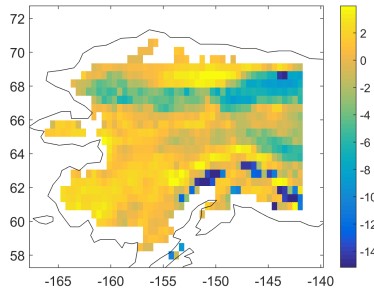

Figure 2: Difference in subsoil temperature (°C) between the models JSBACH and GIPL1.3 from
the University of Alaska Fairbanks (1980-1989 average). JSBACH results from 49.5 m depth.

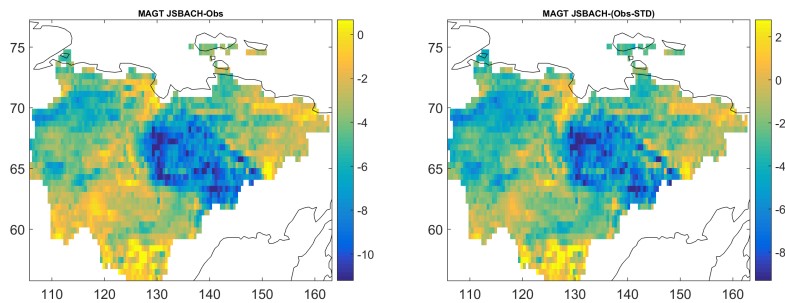

Figure 3: Difference in subsoil temperature (°C) between the JSBACH model (1960-1990 average)
and the geocryological map of Yakutia (Beer et al., 2013). JSBACH results from 49.5 m depth.
The right-hand side figure shows the difference to MAGT mean minus standard deviation from the
geocryological map of Yakutia.





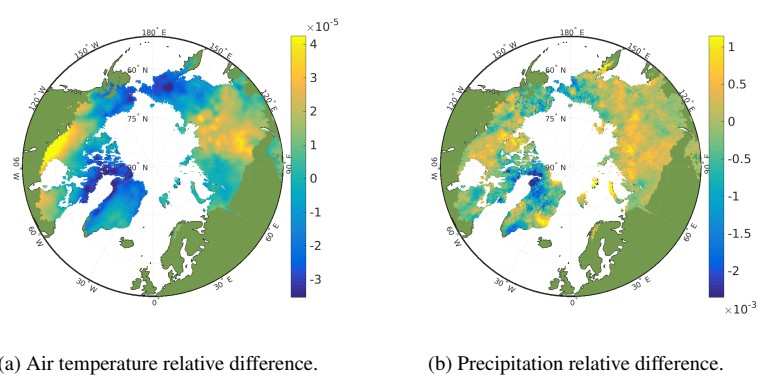

(a) Air temperature relative difference.      (b) Precipitation relative difference.

Figure 4: Comparison of 1980-2009 averages of meteorological variables (REDVAR versus CNTL).

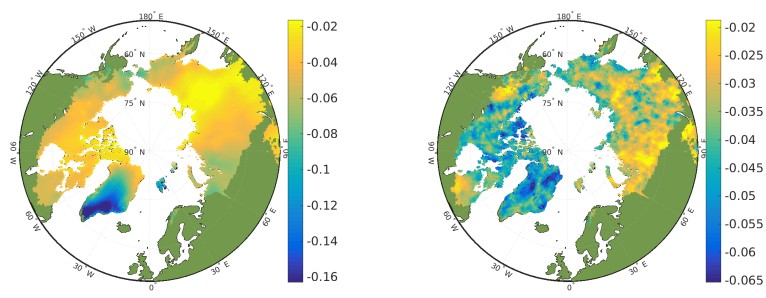

(a) Air temperature standard deviation relative dif- (b) Precipitation standard deviation relative differ-
ference.                                       ence.

Figure 5: Comparison of 1980-2009 standard deviations of meteorological variables (REDVAR ver-
sus CNTL).





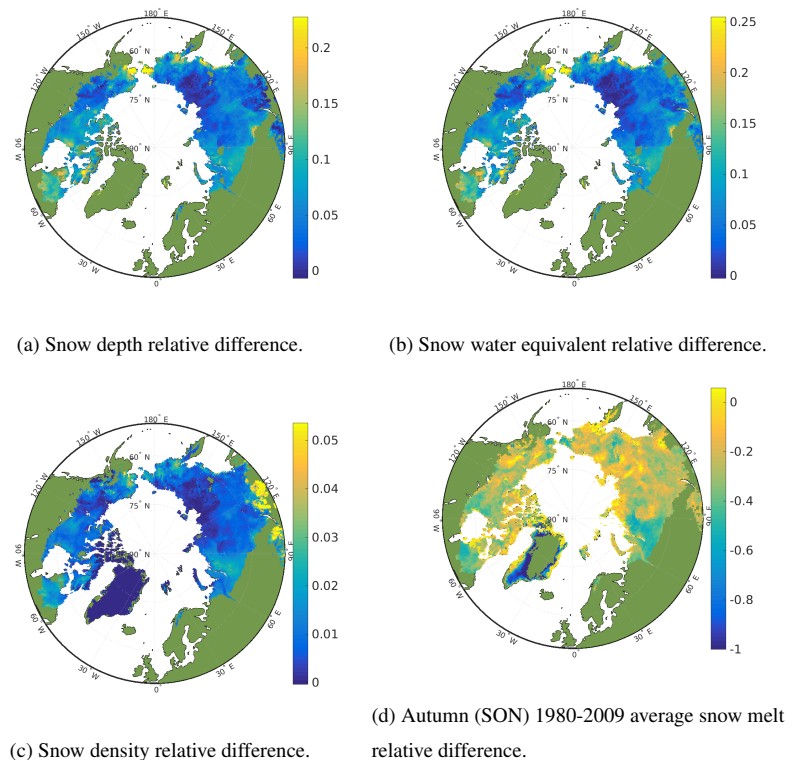

(a) Snow depth relative difference.

(b) Snow water equivalent relative difference.

(c) Snow density relative difference.

(d) Autumn (SON) 1980-2009 average snow melt relative difference.

Figure 6: Comparison of mean winter (DJF) season snow properties during 1980-2009 (REDVAR versus CNTL). Numbers are expressed as a fraction.

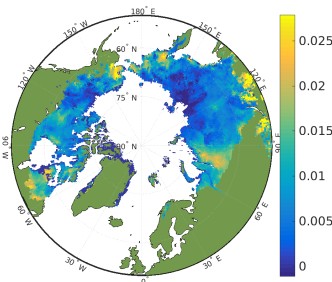

Figure 7: Snow thermal diffusivity relative difference (REDVAR versus CNTL). Numbers are expressed as a fraction.





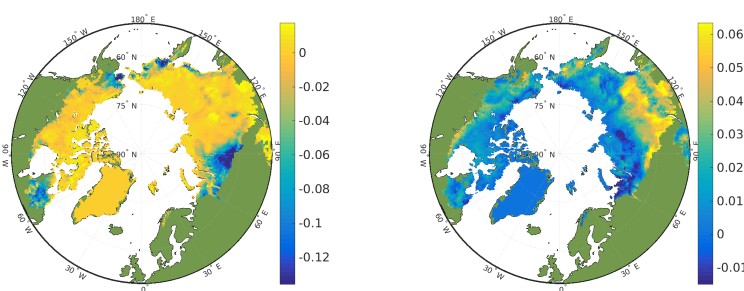

(a) Winter (DJF) lichen and bryophyte thermal diffusivity relative difference.

(b) Summer (JJA) lichen and bryophyte thermal diffusivity relative difference.

Figure 8: Comparison of lichen and bryophyte 1980-2009 average properties (REDVAR versus CNTL). Numbers are expressed as a fraction.

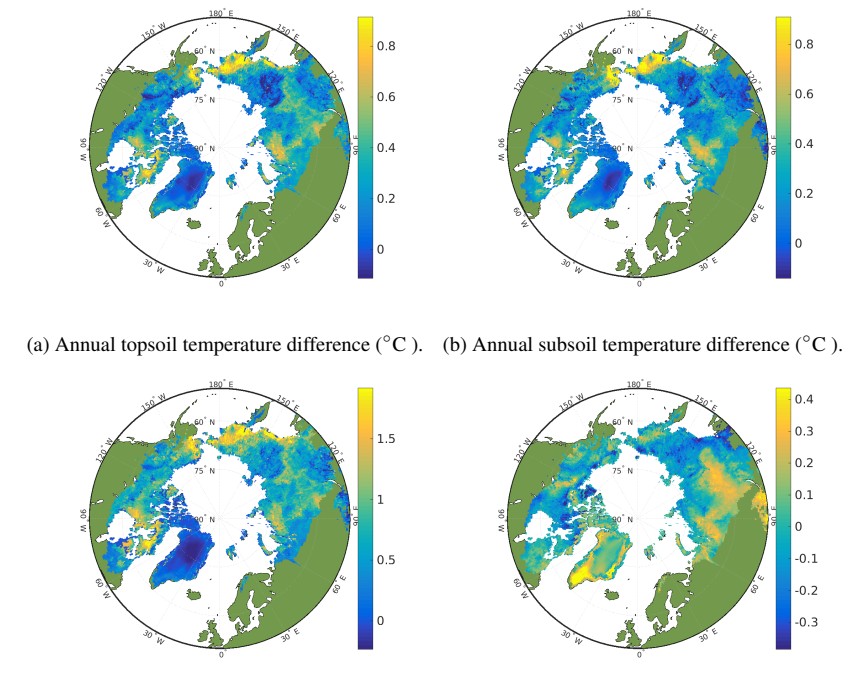

(a) Annual topsoil temperature difference (°C ).

(b) Annual subsoil temperature difference (°C ).

(c) Winter (DJF) topsoil temperature difference (°C ).

(d) Summer (JJA) topsoil temperature difference (°C ).

Figure 9: Comparison of 1980-2009 average soil temperature (REDVAR versus CNTL). Topsoil and subsoil refer to depths of 3 cm and 38 m, respectively.





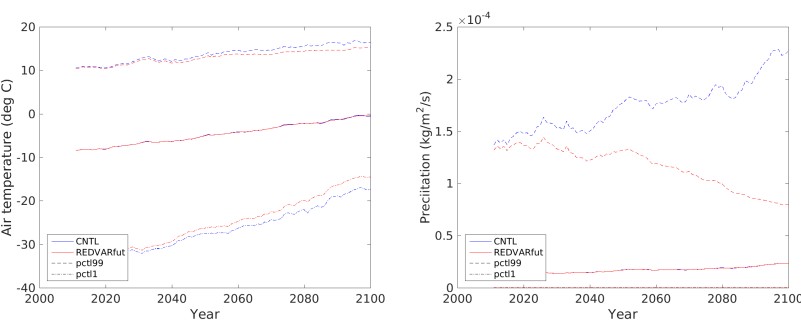

(a) Air temperature (deg C) annual mean, per- (b) Precipitation (kg/m2/s) annual mean, per-
centile 1 and percentile 99. centile 1 and percentile 99.

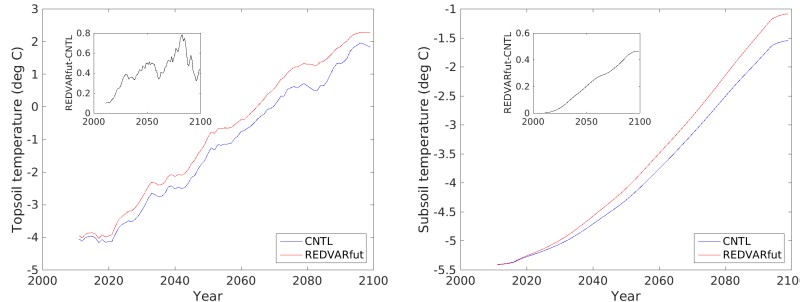

(c) Annual topsoil (3 cm) temperature time series (d) Annual subsoil (38 m) temperature time series
(deg C). 10-year running means are shown. Insets (deg C). 10-year running means are shown. Insets
show the difference time series. show the difference time series.

Figure 10: REDVARfut experiment results at a Canadian site (62.2N/-75.6E) during 2011-2100
showing the effects of changing climate variability on future soil temperature.



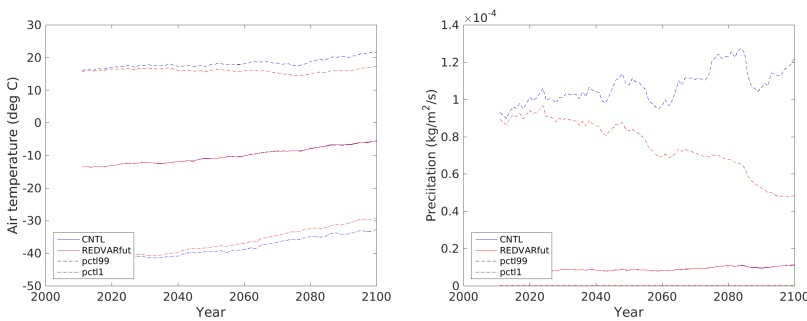

(a) Air temperature (deg C) annual mean, percentile 1 and percentile 99.

(b) Precipitation (kg/m2/s) annual mean, percentile 1 and percentile 99.

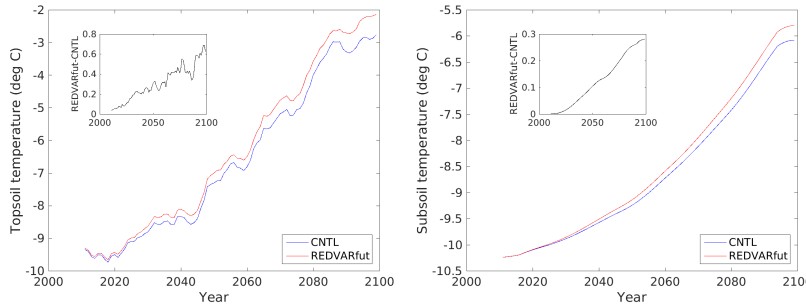

(c) Annual topsoil (3 cm) temperature time series (deg C). 10-year running means are shown. Insets show the difference time series.

(d) Annual subsoil (38 m) temperature time series (deg C). 10-year running means are shown. Insets show the difference time series.

Figure 11: REDVARfut experiment results at a Siberian site (72.2N/147E) during 2011-2100 showing the effects of changing climate variability on future soil temperature.