# Peer review of "Effects of short-term variability of meteorological variables on soil temperature in permafrost regions"

_The Cryosphere, 2017_

## Referee Comment (RC1) · Anonymous Referee #1 · 20 Sep 2017

Effects of short-term variability of meteorological variables on soil temperature in permafrost regions

By Beer et al., 2017.

The study explores short-term variability of the meteorological variables on the permafrost ground temperatures. The major finding is that short-term variability can slow down gradual permafrost warming predicted by many LSM type models. This is mainly true; however, resolution is another important factor that needs to mentioned in the study. For example, at fine scale resolution, topography and vegetation might intercept snow allowing warmer ground temperatures. This effect will not be captured at low res-

olution with global models. Overall, the manuscript is well written and requires some minor edition, which I list below.

L151. Not sure what is reservoir initialization.

REDVAR is introduced in L143 and explained only in L205. Need a better logical flow.

L228. What is the resolution of the GIPL1.3 model? Why GIPL?

L230. Typically, permafrost ground temperature observation made at depth from 15 to 20 m. Why authors choose 38m as comparison depth for the model?

Figures 4 and 9. Since your colorbar include green, I suggest to make non-permafrost areas colorless.

Percentile on the figure introduced in L320, explained in L326. Better flow.

L361. Not sure what authors are trying to say by bolded text.

---

## Referee Comment (RC2) · Anonymous Referee #2 · 11 Nov 2017

The manuscript by Beer et al discusses the role of changes to variability in weather rather than mean climate in governing soil temperature change in the northern high latitudes. The result is that reduced variability is likely to lead to (a) increased snowpack via less frequent melting events, and (b) changed bryophyte thermal conductivity. The authors then take the inverse of this argument to argue that the response under warming is likely to be an increase in the variability, and thus a decrease in snow packs, and thus a reduction in the rate of soil to air warming. The implication of this is that anomaly forcing methods may therefore lead to some biases in the response, although I am not totally clear after reading the paper how important this bias actually is. Nonetheless it is an important point to make and consider.

[Figure]

The article is interesting, well-written, and worth publishing. The basic outline of the argument makes sense, although it would seem simpler if the argument were streamlined to take as a reference case one where variability were held constant and mean values changed in the future transient, and this were compared to the case where both transient means and variabilities were taken from a GCM, as this would not require the change in signs that the current argument requires. I also don't understand why the transient cases were not run globally. This would allow us to more quantitatively estimate the magnitude of the bias as well, rather than having to use the qualitative comparison in the paragraph lines 409-423. So I'd suggest that the authors try to explain why they didn't follow such an approach here.

Minor/specific points:

What are the units in figure 4a ? if unitless, what scale are they relative to – i.e. (T_redvar - T_control) / T_control would be different if in K or C...

The colorbars are really confusing. The same yellow-green-blue colorbar is used throughout, even though each instance of it is being used for a different purpose. I.e. in figure 4, the colorbar is being used for a divergent quantity, but the zero point is in a different point of the colorbar in the two panels. Then in figure 5, the same colorbar is being used for a non-divergent scale, but with two very different gains. Then in figure 6, the same colorbar is used again, but now the zero is at the bottom of the colorscale rather than the top, etc. This sort of thing is really confusing for a reader who must recalibrate their expected color with each new figure. Please try to stick to a convention where you only use one colorbar for one purpose, different types of colorbars for divergent and nondivergent scales, and if you show something with different units, then please use a new and different colorbar. At an absolute minimum, please just keep zero in the same place for each figure.

In figure 6, why isn't the land/ice mask consistent over greenland between the panels?

For figure 10, it looks like the range of variability is reducing quite a bit in the control

case. this is clearly the case with temperature, and most likely the case with precip if expressed relative to the mean value (which, it should be pointed out given the motivation on line 160, is how anomaly forcing for precipitation is applied). and the same appears to be true in figure 11. if this is generally the case, doesn't this undercut the whole argument of the discussion section, which starts out by positing that variability will increase rather than decrease?

For figure 10, how do you compute the 1st and 99th percentiles of annual mean data when you only have a single timeseries that spans a 100-year transient run?

One thing that is missing here that would help the reader assess the importance of the problem is how large is the change in permafrost area, or active layer thickness within permafrost area? Such figures would have required a transient, global reduced variability run. Why couldn't such a run have been produced?

I don't follow the argument on line 436 at all.

---

## Author Comment (AC1) · 8 Dec 2017

**Reviewer #1 comments**

The study explores short-term variability of the meteorological variables on the permafrost ground temperatures. The major finding is that short-term variability can slow down gradual permafrost warming predicted by many LSM type models. This is mainly true; however, resolution is another important factor that needs to mentioned in the study. For example, at fine scale resolution, topography and vegetation might intercept snow allowing warmer ground temperatures. This effect will not be captured at low resolution with global models. Overall, the manuscript is well written and requires some minor edition, which I list below.

*We like to thank the reviewer for their time to carefully read the manuscript and prepare the review. Indeed, your comment on small-scale heterogeneity of soil properties, vegetation and snow depth is very relevant. Since thermal and hydrological processes are nonlinear and heterogeneity is large in these landscapes, we can expect biases when only assuming one average value for state variables, such as temperature or moisture. The interesting question here would be: How would the presented effects of short-term variability of meteorological variables on soil temperature change when also taking into account spatial heterogeneity? At first we can assume little differences:  If temperature goes above freezing point more often, then snow will melt regardless of the actual spatial  differences in snow melt. However, there can be nonlinear threshold effects: If a snowmelt event happens in autumn but there is no snow in some micro-sites, then there will be also no effect of melting. This is an interesting question for further studies. We extended the discussion section of the revised version of the manuscript, 5th paragraph by*

*"Future studies should clarify if these temporal variability effects of meteorological variables on snow depth are lower or higher when additionally taking into account lateral heterogeneity of soil properties (Beer, 2016) or snow, for instance due to snow intercept by topography or vegetation."*

L151. Not sure what is reservoir initialization.

*In transient runs using dynamical models we first need to estimate the state variables at the starting point, which is e.g. the period before the industrialization around 1860. This is done by forcing the dynamic model constantly with the same pre-industrial climate. After some time (decades for physical state variables or millennia for carbon reservoirs) the state variables will not change any more and are in equilibrium with the pre-industrial climate. Now, we can start the transient simulation e.g. until 2010 or into the future with changing climate. This is common practice in Earth system modelling, and here we wanted to explain that we did the full procedure with both climate datasets, CNTL and REDVAR. We however notice that the term "reservoir" is not correct in particular for soil temperature and replace it by "state variables" :*

*"Here, CNTL and REDVAR model runs are done exactly the same way including the spin-up approach for bringing state variables, such as soil temperature in equilibrium with pre-industrial climate."*

REDVAR is introduced in L143 and explained only in L205. Need a better logical flow.

*We agree with this comment and change the order of section 2 subsections such that model experiments are explained after the forcing data explanation.*

L228. What is the resolution of the GIPL1.3 model? Why GIPL?

*The idea was to find an independent soil temperature map for Alaska for evaluating the JSBACH model results. GIPL1.3 uses a semi-analytical solution for mean annual ground temperature following Kudryavtsev, et al., (1974). We believe that this map produced by Sergei Marchenko and Vladimir Romanovsky from the University of Alaska Fairbanks is able to represent large-scale soil temperature pattern in Alaska. The resolution is 2kmx2km which we have further aggregated to 0.5 deg in order to be comparable with JSBACH results. We extend the description of this dataset in the methods section in the following way:*

*"First, JSBACH model results are compared to model results from the GIPL 1.3 model (Sergei Marchenko, University of Alaska Fairbanks) over Alaska for the period 1980-1989. For this we downloaded GIPL model results at 2kmx2km grid cell size from http://arcticlcc.org/products/spatial-data/show/simulated-mean-annual-ground-temperature. Then, the map was reprojected to geographic lat/lon using a bilinear method and further aggreagated to 0.5 degrees grid cell size in order to be comparable with JSBACH outputs."*

L230. Typically, permafrost ground temperature observation made at depth from 15 to 20 m. Why authors choose 38m as comparison depth for the model?

*The observation-based map from the Melikov Permafrost Institute (Fedorov et al., 1989, 1991) presents permafrost temperature and we assume it represents mean annual ground temperature which is thought to be constant over time. In order to be comparable with this information, we chose the last soil layer of model results.*

*However, we notice a small mistake in the figure 2 and 3 captions and replace 49.5 by 38 m depth.*

*For the evaluation using borehole measurements (Fig 1), however, we were using particularly model results from the same depth as the measurements. This is now further clarified by extending the figure 1 caption by the sentence*

*"Model results are taken from the depth of observation for each point."*

Figures 4 and 9. Since your colorbar include green, I suggest to make non-permafrost

areas colorless.

*We agree with this and in the revised version of the manuscript use white for non-land pixels and gray for non-permafrost land pixels. Please, also note all the changes in color bar and color scale in these plots according to the comments of reviewer 2.*

Percentile on the figure introduced in L320, explained in L326. Better flow.

*The information on percentiles used for minimum and maximum values is repeated at both locations in the text now.*

L361. Not sure what authors are trying to say by bolded text.

*In the results section we compare the reduced climate variability JSBACH results (REDVAR) with the control variability JSBACH results (CNTL) while in the discussion section we address the question on the effects of potentially increasing variability in future.*

**Reviewer #2 comments**

The manuscript by Beer et al discusses the role of changes to variability in weather rather than mean climate in governing soil temperature change in the northern high latitudes. The result is that reduced variability is likely to lead to (a) increased snowpack via less frequent melting events, and (b) changed bryophyte thermal conductivity. The authors then take the inverse of this argument to argue that the response under warming is likely to be an increase in the variability, and thus a decrease in snow packs, and thus a reduction in the rate of soil to air warming.

The implication of this is that anomaly forcing methods may therefore lead to some biases in the response, although I am not totally clear after reading the paper how important this bias actually is. Nonetheless it is an important point to make and consider.

*We thank the reviewer for their time to read and carefully evaluate this manuscript. First of all, we learn about a potential lower future soil warming due to increasing climate variability but you are right that a second implication is the importance of considering higher statistical moments in bias correction methods, as pointed out in the conclusion section. One additional conclusion from this study is that snow and near-surface vegetation functions are essential for the land-atmosphere heat flux and hence need to be represented in Earth System Models.*

The article is interesting, well-written, and worth publishing. The basic outline of the argument makes sense, although it would seem simpler if the argument were streamlined

to take as a reference case one where variability were held constant and mean values changed in the future transient, and this were compared to the case where both transient means and variabilities were taken from a GCM, as this would not require the change in signs that the current argument requires.

*Thank you for this additional idea which indeed could be interesting to look at in a separate study. Following your suggestion, we would have two climate datasets that show a difference in short-term variability while long-term averages are conserved.*

*Beside long-term and annual-seasonal variability, there are also different characteristic patterns of inter-monthly, inter-weekly or inter-daily variability (Mahecha et al., 2010). These can also greatly differ between ecosystems and even in time. Even with carefully producing such dataset of constant short-term variability, still there is a risk of unintentionally producing unrealistic climate for the specific location, e.g. there is uncertainty in estimating the characteristic variabilities. It is also unclear if the characteristic short-term variabilities should be derived at annual or seasonal or monthly scale? This is an interesting idea for future studies but required a lot of more detailed and careful thinking and data preparation. With our method we keep as much as possible the characteristics the original dataset and just reduce the variability of residuals to the mean seasonal cycle. Our idea was to generate an as realistic as possible climate data for the period 1901-2010 also in the artificial REDVAR dataset in order to avoid additional indirect artificial effects on ecosystem states and functions (JSBACH results).*

*Still, it would be interesting to reduce the variability of future projections as is discussed below. We already did this for several individual grid cells, but we see a major limitation due to manpower, disk storage and CPU time for producing such dataset and running the JSBACH model at pan-Arctic scale.*

I also don't understand why the transient cases were not run globally. This would allow us to more quantitatively estimate the magnitude of the bias as well, rather than having to use the qualitative comparison in the paragraph lines 409-423. So I'd suggest that the authors try to explain why they didn't follow such an approach here.

*In a major effort we designed the global REDVAR dataset 1901-2010 with constantly reduced short-term variability of climate variables and performed the CNTL and REDVAR model runs. The assumption is that comparing the results from these two model experiments will provide the same insight as ncreasing the variability continuously from 2010 until 2100. To evaluate this assumption we repeated the whole procedure for 2 grid cells: 1) design the REDVARfut dataset with increasing variability during 2010-2100, and ii) perform the JSBACH runs. Indeed, results from these additional runs are very similar to the global runs using the constant difference in variability.*

*In general, one could also repeat the whole study and design a REDVAR climate dataset with increasing reduced variability until 2100 and again force JSBACH with that climate in addition to a CNTL. However, manpower, storage capacity and CPU time available at the moment are the*

*limiting factors. Please, keep in mind that if future projections are the focus of the study, the whole procedure including REDVAR data preparation is required for an ensemble of climate model results and different RCP scenarios. From the results at the two locations in Canada and Siberia, which were conducted in addition to the main modelling experiment at continental scale, we are however confident that our main conclusions would not change.*

*Please, see our suggestion for extending the discussion in this respect as presented in the response to your question on permafrost extend and active-layer thickness below.*

Minor/specific points:

What are the units in figure 4a ? if unitless, what scale are they relative to – i.e. (T_redvar - T_control) / T_control would be different if in K or C...

*This is actually a very good point. All temperatures in this manuscript are in degree Celsius. We improve the figure captions in order to make underlying units clear. If it was Kelvin in Fig 4a, then the relative difference of these mean values would have been even much smaller when dividing the difference by several hundreds of Kelvin. We decided now to show the absolute difference of temperature averages in degree C as also done for the soil temperature results. The main point of that figure is to see that indeed, long-term averages are similar between the datasets. The color scale of Fig 4a is now adjusted to match that one of Fig 9 in order to address the next comment.*

The colorbars are really confusing. The same yellow-green-blue colorbar is used throughout, even though each instance of it is being used for a different purpose. I.e. in figure 4, the colorbar is being used for a divergent quantity, but the zero point is in a different point of the colorbar in the two panels. Then in figure 5, the same colorbar is being used for a non-divergent scale, but with two very different gains. Then in figure 6, the same colorbar is used again, but now the zero is at the bottom of the colorscale rather than the top, etc. This sort of thing is really confusing for a reader who must recalibrate their expected color with each new figure. Please try to stick to a convention where you only use one colorbar for one purpose, different types of colorbars for divergent and nondivergent scales, and if you show something with different units, then please use a new and different colorbar. At an absolute minimum, please just keep zero in the same place for each figure.

*Thank you very much for this comment. All color bars and color scales have been adjusted accordingly. With this we hope to improve the readability.*

*Our reasoning is as follows: A) Differences between mean temperatures are displayed using one constant color bar and color scale. (Fig 4a and Fig 9). B) Effects on snow (Fig 6) or moss (Fig 8) properties get another constant color bar and color scale each. For this purpose we swapped snow thermal diffusivity and snow melt result figures (Fig 6d and Fig 7) and carefully adjusted the figure references in the text. C) All other maps come with their own color bar and color scale. D) Exceptions are Fig 2 and 3 (model evaluation) for which we use the same color bar than for Fig 6*

*but the maps are really different (region and projection). Therefore we assume that using the same color bar here will not confuse the reader. Still, color scales for Fig 3a and 3b are adjusted.*

*Due to the standardized color scales according to purposes, we cannot see any spatial details in Fig 4a anymore. That is logical and reflects the whole message of the study: mean air temperature differences are negligible while mean annual ground temperature differ by up to 0.8 degree C due to the difference in climate variables standard deviation.*

In figure 6, why isn't the land/ice mask consistent over greenland between the panels?

*These differences were just due to the fact that the model assumes constant glacier cover in these areas but the way snow and soil processes are treated differ between glacier and land areas in JSBACH. Since glaciers are not the focus of this study, plotting glacier regions is therefore not useful and in the improved plots all glacier area results are removed.*

*During that work we also recognized that for a few locations for which we do have borehole measurements from GTN-P the model grid cell assumes glacier cover. Therefore, these locations are also not useful for the Fig 1 model evaluation exercise and hence removed in this updated version of the manuscript. The model-data mismatch is now further reduced and the whole conclusion from the evaluation exercise remains.*

For figure 10, it looks like the range of variability is reducing quite a bit in the control case. this is clearly the case with temperature, and most likely the case with precip if expressed relative to the mean value (which, it should be pointed out given the motivation on line 160, is how anomaly forcing for precipitation is applied). and the same appears to be true in figure 11. if this is generally the case, doesn't this undercut the whole argument of the discussion section, which starts out by positing that variability will increase rather than decrease?

*Thank you very much for this comment which let us recognize that the applied percentiles were not an ideal metric for the purpose of characterizing the difference in the data. We are not interested in the 1-2 most extreme days during the year and if their temperature or precipitation changed in time, but in the day-to-day and inter-weekly variability of meteorological variables. Therefore, we now compute the mean absolute difference (MAD) between daily data of both datasets during each year and show 10-year running means of these annual time series in the insets of Fig 10 and 11 a and b. Here, we can see that this difference in the daily data between the datasets is increasing with time while the mean annual data remains similar. The MAD definition has been added to the methods section 2.6, equation 5.*

*Still, your comment on the variability of the control dataset is valid. To address this question, the following plot shows the mean absolute deviation (difference of daily data to the mean) using a 1-year moving window approach. However, the mean seasonal cycle has been removed from the daily data before in that analysis because we are only interested in the short-term variability and not in changing mean seasonal cycles. One can see that this short-term variability is decreasing at*

[Figure]

*the one example location in Canada. (Still, the REDARfut data would decrease even more according to the inset of Fig 10a).*

*Therefore, you are right and there will be grid cells of decreasing climate variability in Arctic regions in our CNTL dataset. Your question was if that will undercut the motivation of the study. That is clearly not the case. In our study we are using one specific climate dataset (bias-corrected MPI-ESM output from the CMIP5 archive, see section 2.2) which is not representative for an ensemble of climate model results as e.g. used in Seneviratne et al., (2012). We are just using this dataset as CNTL in order to compare JSBACH model results to the artificially reduced variability dataset (REDVARfut) from which we then learn about the effects of climate variability on mean soil temperature. These additional model runs into the future at two grid cells are also only complementing the main experiment presented in this paper at a continental scale for the period until 2010, and the results in Fig 10 and 11 confirm the continental scale results in Fig 9. We therefore do not expect a different result if the CNTL dataset showed an increasing short-term variability instead.*

For figure 10, how do you compute the 1st and 99th percentiles of annual mean data when you only have a single timeseries that spans a 100-year transient run?

*We calculated percentiles of daily data for each year but as stated in the above response these percentiles have not been a useful measure for the change in variability and we replaced them by the mean absolute difference between daily data during a year, see comment above, equation 5 and Fig 10 and 11.*

One thing that is missing here that would help the reader assess the importance of the problem is how large is the change in permafrost area, or active layer thickness within permafrost area? Such figures would have required a transient, global reduced variability run. Why couldn't such a run have been produced?

*We fully agree with the reviewer in this is indeed a very interesting question. One could address this question with new pan-Arctic scale model experiments until 2100 or 2300 for which we would also need to process new climate forcing data of reduced variability. Please, see our response above to this point. The main limitation here is CPU time and disk storage; it would be a whole project in its own over the next year.*

*Actually, a 0.2 to 0.8 °C cooler soil due to increasing climate variability can have a significant effect on future permafrost area extend. This can be seen when looking into recent pan-Arctic projections of soil temperature under different warming scenarios. For instance in Schaphoff et a. (2013) future soil temperature at 38 cm depth is projected to reach -1 to +1 °C over the major part of the current permafrost area depending on the warming scenario. For a mean annual ground temperature close to the freezing point, a difference of 0.5 °C will matter.*

*We extend the discussion section, second last paragraph by*

*"Soil temperature is projected to arrive at values around the freezing point in 38 cm depth over the major part of the current permafrost area (Schaphoff et al., 2013). Therefore, differences of soil temperature of 0.1 to 0.8 °C due to changing climate variability would have an effect on active-layer thickness and permafrost extent, too. It would be interesting to generate an additional artificial REDVARfut dataset with pan-Arctic cover and investigate in detail the impacts of climate variability on active-layer thickness and permafrost extend at the end of the century in a future project."*

I don't follow the argument on line 436 at all.

*This sentence was isolated within the manuscript and not useful. It has been removed it in the revised version of the manuscript.*

**References**

Fedorov, A. N., Botulu, T. A., and Varlamov, S. P.: Permafrost Landscape of Yakutia (in Russian), Novosibirsk: GUGK, 1989.

Fedorov, A. N., Botulu, T. A., and Varlamov, S. P.: Permafrost Landscape Map of Yakutia ASSR, Scale 1:2500000, Moscow: GUGK, 1991.

Kudryavtsev VA, Garagulya LS, Kondrat'yeva KA, Melamed VG (1974) *Fundamentals of Frost Forecasting in Geological Engineering Investigations (in Russian)*. Nauka, Moscow.

Mahecha, M. D., et al. (2010) Comparing observations and process-based simulations of biosphere-atmosphere exchanges on multiple timescales, J. Geophys. Res., 115, G02003, doi:10.1029/2009JG001016.

Schaphoff, S., Heyder, U., Ostberg, S., Gerten, D., Heinke, J., and Lucht,W.: Contribution of permafrost soils to the global carbon budget, Environmental Research Letters, 8, 014 026, http://iopscience.iop.org/1748-9326/8/1/014026, 2013

Seneviratne, 630 S., Nicholls, N., Easterling, D., Goodess, C., Kanae, S., Kossin, J., Luo, Y., Marengo, J., McInnes, K., Rahimi, M., Reichstein, M., Sorteberg, A., Vera, C., and Zhang, X.: Changes in climate extremes and their impacts on the natural physical environment, in: Managing the Risks of Extreme Events and Disasters to Advance Climate Change Adaptation, edited by Field, C., Barros, V., Stocker, T., Qin, D., Dokken, D., Ebi, K., Mastrandrea, M., Mach, K., Plattner, G., Allen, S., Tignor, M., and Midgley, P., pp. 109–230, A Special Report of Working Groups I and II of the Intergovernmental Panel on Climate Change (IPCC). Cambridge University Press, Cambridge, UK, and New York, NY, USA, 2012.

---

## Author Response (AR2)

Page 7:
- "frost"-enhanced JSBACH model (page 7)
I suggest to replace "frost" with permafrost or "freeze thaw".

**We are using permafrost now.**

- http://arcticlcc.org/products/spatialdata/show/simulated-mean-annual-ground-temperature
Please provide a citation if available.

**There was no reference given to the dataset but we included this citation which should be the right reference to GIPL 1.3.**

**Marchenko, S.; Romanovsky, V. & Tipenko, G. Numerical modeling of spatial permafrost dynamics in Alaska Proceedings of the Ninth International Conference on Permafrost, Fairbanks, Alaska, USA, 29 June–3 July 2008.**

-"..respective GTN-P Thermal State of Permafrost (TSP) snapshot data has been dowloaded from the National Snow and Ice Data Center (NSIDC)…"
Please provide the link and potential citation.

**We believe it is important to cite the scientific papers behind this compilation of borehole measurements: Romanovsky et al., 2010; Christiansen et al., 2010; Smith et al., 2010. In addition we further included URL and citation of the specific dataset that has been downloaded from the NSIDC:**

**"The respective GTN-P Thermal State of Permafrost (TSP) snapshot data (International Permafrost Association (IPA), 2010) has been dowloaded from the National Snow and Ice Data Center (NSIDC) at http://nsidc.org/data/G02190#"**

- You are using °C and deg C, please unify throughout the paper

**Many thanks, that is corrected.**

Page 10:
-"Under these climate conditions, it is raining more often a little bit and air temperature are not extreme resulting in more moist conditions for lichens and bryophytes, hence higher thermal diffusivity."
Please correct sentence (air temperature is)

**Thank you**

Page 11:
- The increasing differences in the variability of meteorological variables under conserved long-term averages leads
Please correct sentence (lead)

**Thank you**

Page 12:
- "the direction of the conclusions are carefully inverted in this discussion section."
Please rephrase, it is not clear what this sentence means (as already suggested by the reviewer).
Why is this text bolded and highlighted? I suggest to remove this format.

**Bold text format has been removed. The sentence has been revised as follows; hopefully better understandable:**

**"However, for interpreting the results in terms of future ecosystem responses to {\itshape increasing} climate variability \cite[]{Seneviratne2012}, the results of the CNTL model run are compared against the results of the REDVAR model run in this discussion section (CNTL-REDVAR)."**

**For the discussion it makes more sense to discuss what would be the effect of increasing variability and hence treat REDVAR as the control model run.**

Page 22: Figure 4: Please correct format (a and b should be aligned)?

**This is corrected**

Page 23: Figure 6: Please align figures.
**This is corrected**

None of your figures have a- d labeling, but captions refer to a-d. Please add.
**The labelling of subfigures is done automatically by latex in the sub-captions. We hope this is in line with publisher and journal specifications.**

Page 14: Conclusion
-Overall, the soil temperature response to increasing climate variability and extreme event frequency (soil cooling) will be opposite to the response of soil temperature to gradually increasing air temperature (soil warming). This shows the importance of representing dynamically snow and lichen and bryophyte functions in Earth system models for projecting future permafrost soil states and land-atmosphere interactions, hence future climate.

I do not understand why these sentences are connected ("This shows.."). Please add an explanatory sentence connecting these two sentences.

**We restructured (one sentence moved) and slightly rephrased the conclusions section as follows. Hopefully, this is better understandable now:**

**"Artificial model experiments have been used in order to quantify the impact of the variability of meteorological variables on the long-term mean of mean annual ground temperature in permafrost-affected terrestrial ecosystems. In future, the soil temperature response to increasing climate variability and extreme event frequency (soil cooling) will be opposite to the response of soil temperature to gradually increasing air temperature (soil warming). Is has been shown that snow and near-surface vegetation dynamics are the underlying mechanisms for this. Therefore, dynamics of snow and lichen and bryophyte functions need to be represented in Earth system models for validly**

projecting future permafrost soil states and land-atmosphere interactions, hence future climate. Our findings also point to the need to represent changes in short-term variability of meteorological variables in bias-corrected climate data of future periods."